# Enzymatic Properties of Recombinant Phospho-Mimetic Photorespiratory Glycolate Oxidases from *Arabidopsis thaliana* and *Zea mays*

**DOI:** 10.3390/plants9010027

**Published:** 2019-12-24

**Authors:** Mathieu Jossier, Yanpei Liu, Sophie Massot, Michael Hodges

**Affiliations:** Institute of Plant Sciences Paris-Saclay, CNRS, Université Paris-Sud, INRA, Université d’Evry, Université Paris-Diderot, Université Paris-Saclay, 91405 Orsay CEDEX, France; mathieu.jossier@ips2.universite-paris-saclay.fr (M.J.); sophie.massot@ips2.universite-paris-saclay.fr (S.M.)

**Keywords:** *Arabidopsis thaliana*, glycolate oxidase, photorespiration, protein phosphorylation, *Zea mays*

## Abstract

In photosynthetic organisms, the photorespiratory cycle is an essential pathway leading to the recycling of 2-phosphoglycolate, produced by the oxygenase activity of ribulose-1,5-bisphosphate carboxylase/oxygenase, to 3-phosphoglycerate. Although photorespiration is a widely studied process, its regulation remains poorly understood. In this context, phosphoproteomics studies have detected six phosphorylation sites associated with photorespiratory glycolate oxidases from *Arabidopsis thaliana* (*At*GOX1 and *At*GOX2). Phosphorylation sites at T4, T158, S212 and T265 were selected and studied using Arabidopsis and maize recombinant glycolate oxidase (GOX) proteins mutated to produce either phospho-dead or phospho-mimetic enzymes in order to compare their kinetic parameters. Phospho-mimetic mutations (T4D, T158D and T265D) led to a severe inhibition of GOX activity without altering the K_M_ glycolate. In two cases (T4D and T158D), this was associated with the loss of the cofactor, flavin mononucleotide. Phospho-dead versions exhibited different modifications according to the phospho-site and/or the GOX mutated. Indeed, all T4V and T265A enzymes had kinetic parameters similar to wild-type GOX and all T158V proteins showed low activities while S212A and S212D mutations had no effect on *At*GOX1 activity and *At*GOX2/*Zm*GO1 activities were 50% reduced. Taken together, our results suggest that GOX phosphorylation has the potential to modulate GOX activity.

## 1. Introduction

Photorespiration begins with the fixation of O_2_ to ribulose-1,5-bisphosphate by ribulose-1,5-bisphosphate carboxylase/oxygenase (RuBisCO) leading to the formation of one molecule of 2-phosphoglycolate (2PG) and one molecule of 3-phosphoglycerate (3PGA). 2PG is then metabolized to produce 3PGA by the photorespiratory cycle which occurs in four subcellular compartments (chloroplasts, peroxisomes, mitochondria and cytosol), and involves eight core enzymes and several transporters [1]. Photorespiration has a negative impact on plant yield since it limits photosynthetic CO_2_ assimilation due to competition at the RuBisCO active site, and it releases assimilated carbon and nitrogen as CO_2_ and ammonium that have to be either reassimilated at an energetic cost or lost. This has led to efforts to minimize the negative effects of photorespiration to improve plant yield by producing plants containing a chloroplastic bypass to metabolize photorespiratory glycolate (the most recent examples being [2,3]). However, photorespiratory glycolate produced in the chloroplast from toxic 2PG [4] is normally metabolized in peroxisomes by glycolate oxidase (GOX), a flavin mononucleotide (FMN) containing enzyme that catalyzes the transformation of glycolate to glyoxylate with the production of hydrogen peroxide [5]. This enzyme evolved from a bacterial lactate oxidase and it is a member of the α-hydroxy-acid oxidase superfamily [6]. In *Arabidopsis*, there are five GOX-related genes: *At3g14420*, *At3g14415*, *At4g18360*, *At3g14130* and *At3g14150* (encoding *At*GOX1, *At*GOX2, *At*GOX3, *At*HAOX1 and *At*HAOX2, respectively). According to transcriptomic analyses, *At3g14415* (*AtGOX2*) and *At3g14420* (*AtGOX1*) are highly expressed in leaves and they represent the major GOX isoforms in *Arabidopsis thaliana* [7]. *AtGOX3* is mainly expressed in senescing leaves and roots, where it has been proposed to also function as a lactate oxidase and thus play a role in lactate metabolism [8]. *AtHAOX1* and *AtHAOX2* are expressed in seeds and encode proteins preferring medium- and long-chain hydroxyl acids as substrates [6]. Knock-out mutants of each Arabidopsis *GOX* gene do not exhibit a photorespiratory growth phenotype although they were all more sensitive to *Pseudomonas syringae* and to ozone [9,10]. A photorespiratory phenotype in air that was reversed by elevated CO_2_ (3000 ppm) was observed however in an Arabidopsis artificial miRNA GOX line (*amiRgox1/2*) with both *AtGOX1* and *AtGOX2* knocked-down and only 5%–10% of wild-type Arabidopsis leaf GOX activity [7]. The transfer of *amiRgox1/2* plants from high CO_2_ to ambient air led to a 700-fold accumulation of glycolate and a reduced carbon allocation to sugars, organic acids and amino acids that induced an early senescence of older leaves [7]. Even though *At*GOX1 and *At*GOX2 showed a redundant photorespiratory function, *At*GOX1 appeared to have a more predominant role in photorespiration since it attenuated the phenotype of a *cat2* mutant [11]. The importance of GOX has been observed also in tobacco, rice and even maize (a C4-plant), where a reduction of GOX activity in RNAi, antisense or mutant lines led to delayed growth in air associated with a decrease of net CO_2_ assimilation rate [12,13,14,15].

Since photorespiration interacts with several metabolic processes including photosynthesis, nitrogen metabolism, respiration, C1 metabolism as well as H_2_O_2_ production by GOX [16], it might be expected that the photorespiratory cycle would be coordinated with these processes and modulated according to metabolic needs and perhaps environmental cues. Even though photorespiration has been widely studied over the last few decades, regulation of this C2-cycle and its enzymes is still poorly understood. It has been proposed that serine acts as a signal to regulate the expression of photorespiratory genes [17]. Several peroxisomal photorespiratory enzymes have been associated with putative post-translational modifications such as ubiquitination, nitration, persulfidation, and acetylation (for a review see [18]). Indeed, the activity of pea GOX was found to be inhibited by S-nitrosylation [19] as was Arabidopsis glycine decarboxylase [20]. The glycerate kinase of maize was shown to be redox-regulated by thioredoxin f; however, this was not the case for the Arabidopsis enzyme [21]. Arabidopsis mitochondrial glycine decarboxylase L-protein activity was found to be redox regulated by thioredoxin [22,23]. Protein phosphorylation could be another post-translational mechanism involved in the regulation of the photorespiratory cycle as phosphoproteomics studies have identified a number of phosphopeptides associated with all but one of the core photorespiratory enzymes [24]. Concerning photorespiratory *At*GOX1 and *At*GOX2, several phosphorylation sites (T4, T155, T158, S212, T265 and T355) have been reported (see Table 1).

In this study, the consequences of T4, T158, S212 and T265 phosphorylation on *At*GOX1 and *At*GOX2 enzymatic activities and kinetic properties were analysed using purified recombinant wild-type, phospho-dead and phosphorylation-mimetic GOX proteins. This was also carried out to test the equivalent residues (T5, T159, S213 and T266) of maize photorespiratory GOX (*Zm*GO1) [15]. Our results are discussed in terms of the possible consequences of each GOX phosphorylation on enzyme structure and function and in response to environmental stresses and future avenues to explore to better understand photorespiratory GOX phosphorylation are proposed.

## 2. Results

### 2.1. Phosphopeptides and Phosphorylated Residues of Arabidopsis GOX1 and GOX2

In the context of a possible regulation of photorespiratory GOX by protein phosphorylation, a first step was to retrieve potential phosphorylated residues from phosphopeptides associated with *At*GOX1 and *At*GOX2 from published phosphoproteomics studies and the PhosPhAt 4.0 database (http://phosphat.mpimp-golm.mpg.de/) (see Table 1). In this way, six phosphopeptides were found with six different phosphorylated residues; T4, T155, T158, S212, T265 phosphopeptides associated with both *At*GOX1 and *At*GOX2 and T355 that was in a peptide associated only with *At*GOX1 (Table 1).

All of these phosphorylated amino acids were conserved among *At*GOX1, *At*GOX2 and *At*GOX3 proteins while T4 and T158 were replaced by a valine in *At*HAOX1 and *At*HOAX2 (Appendix A). To analyse the conservation of these phosphorylated residues amongst (S)-2-hydroxy-acid oxidases (EC 1.1.3.15) of bacteria, cyanobacteria, algae, human and plants, we compared amino acid sequences of lactate oxidase (LOX) from *Aerococcus viridians, Lactococcus lactis, Nostoc* and *Chlamydomonas reinhardtii,* GOX from *Zea mays*, *Arabidopsis thaliana*, *Brassica napus*, *Vitis vinifera*, *Nicotiana benthamiana*, *Spinacia oleracea*, *Populus alba* and *Homo sapiens* and HAOX from *Arabidopsis thaliana* and *Homo sapiens* (Appendix A). The residues corresponding to T155, S212 and T265 of Arabidopsis GOX were conserved as either a serine or a threonine in all species (except for T265 in *Chlamydomonas reinhardtii* LOX, S212 in *Lactococcus* LOX and HAOX2 of *Homo sapiens,* T265 in *Chlamydomonas reinhardtii* LOX), whereas T4 was conserved only in the plant GOX proteins while T158 was conserved in all GOX proteins, human HAOX2, *Aerococcus viridians* LOX and *Lactococcus lactis* LOX (Appendix A). Based on these observations, it was decided to investigate further the phosphoregulation of photorespiratory GOX by characterising phospho-mimetic recombinant *At*GOX1 and *At*GOX2 as well as the C4-plant enzyme, *Zm*GO1 that had been shown to be important for growth in air (400 ppm CO_2_) [15]. Among the six phosphorylation sites, the highly conserved plant GOX residues, T4, T158, S212 and T265 of *At*GOX1 and *At*GOX2 were chosen for this study that corresponded to T5, T159, S213, T266 of *Zm*GO1 (Appendix A).

### 2.2. Phospho-Mimetic AtGOX1, AtGOX2 and ZmGO1 Exhibit Altered Glycolate Oxidase Activities

To investigate the potential regulation of *At*GOX1 and *At*GOX2 activity by protein phosphorylation and to see if this was conserved in maize photorespiratory GO1, the kinetic parameters of purified recombinant N-terminal His-tagged GOX proteins (see Appendix A) was undertaken. All selected phosphorylated residues were replaced by an aspartate to mimic a constitutive phosphorylation. To produce phospho-dead GOX proteins, S212/213 and T265/266 (Arabidopsis GOX/*Zm*GO1 numberings) were replaced by an alanine while T4/5 and T158/159 were changed to a valine to also mimic the sequence of *At*HOAX1 and *At*HOAX2 (and *Homo sapiens* HAOX2 for the T4 position) (Appendix A) so as to evaluate the role of these amino acids in substrate specificity. Using glycolate as a substrate, *At*GOX1, *At*GOX2 and *Zm*GO1 presented rather similar K_M_ glycolate (210 µM, 279 µM and 126 µM) and k_cat_ (11.12 s^−1^, 10.93 s^−1^ and 14.45 s^−1^) values (Table 2). The T4/5V mutations did not have any consequences on either K_M_ glycolate or k_cat_ while the T4/5D phospho-mimetic mutations drastically decreased (by 10–20 fold) the k_cat_ of the three recombinant GOX enzymes without significantly altering the Km glycolate (Table 2). Similar results were observed for the T265/266 phospho-site since its mutation to alanine did not have any effect on the calculated kinetic parameters while k_cat_ was strongly decreased for *At*GOX1_T265D_, *At*GOX2_T265D_ and *Zm*GO1_T266D_ proteins by 99%, 81% and 95% (Table 2). *At*GOX1_T158V_, *At*GOX2_T158V_ and *Zm*GO1_T159V_ also showed altered kinetic parameters with an improved K_M_ glycolate (2–3 fold lower) but a 4–5 fold decreased k_cat_ compared to their wild-type GOX counterparts (Table 2). When this residue was mutated to mimic a phosphorylated GOX, the resulting recombinant proteins (*At*GOX1_T158D_, *At*GOX2_T158D_ and *Zm*GO1_T159D_) were inactive (Table 2). Perhaps surprisingly, the mutated S212/213 phospho-site gave a differential affect amongst the three GOX proteins studied. While *At*GOX1_S212A_ and *At*GOX1_S212D_ did not show any differences in their kinetic parameters compared to *At*GOX1_WT_, both *At*GOX2_S212A_ and *At*GOX2_S212D_ as well as *Zm*GO1_S213A_ and *Zm*GO1_S213D_ exhibited an approximately 2-fold decrease of their k_cat_ with no change in K_M_ glycolate (Table 2).

### 2.3. Phospho-site Mutations at T4/5 and T158/159 Have a Limited Effect on Substrate Specificity

*At*HOAX1 and *At*HAOX2 proteins preferentially use long-chain hydroxy-acids as substrates although *At*HOAX1 can also quite efficiently use both lactate and glycolate [6]. Since the two Arabidopsis HAOX enzymes had a valine at the T4 and T158 positions, it was decided to test whether the mutation of these residues could give rise to substrate-specific effects. To achieve this, activity measurements were repeated with either L-lactate or 2-hydroxyoctanoate using T4/5 and T158/159 phospho-site mutated GOX proteins and their kinetic parameters were calculated and compared (Table 3). We chose to test 2-hydroxyoctanoate because it was found to be a good substrate for both Arabidopsis HAOX enzymes [6]. First, however, the substrate specificity of GOX_WT_ proteins were compared and as previously reported, the Km L-lactate was higher than the Km glycolate [8] with an approximate 8-fold higher value for *At*GOX1 and *At*GOX2 while only a 4-fold difference was seen for *Zm*GO1, and surprisingly, the calculated k_cat_ values for the lactate oxidase reaction were not significantly different when compared to the glycolate oxidase activities (Table 2; Table 3). When using 2-hydroxyoctanoate as a substrate, again a higher Km was observed compared to glycolate for *At*GOX1 and *At*GOX2 (3.6-fold and 1.8-fold, respectively), while the Km 2-hydroxyoctanoate for *Zm*GO1 remained unchanged (Table 2 and Table 3). However, the k_cat_ for the 2-hydroxyoctanoate reaction was halved for all recombinant GOX_WT_ proteins when compared to glycolate oxidase activity (Table 2 and Table 3). Taken together, these results showed that our recombinant GOX_WT_ proteins were more efficient (based on k_cat_/Km) using glycolate as a substrate, although *Zm*GO1 appeared to be less selective.

The effect of the selected (T4/5 and T158/159) phospho-site mutations on the calculated kinetic parameters of the lactase oxidase and 2-hydroxyoctanoate oxidase reactions was compared. It was found that T158V/D and T159V/D mutations led to similar effects on K_M_ and k_cat_ (T158/159V) and activity (T158/159D) using either L-lactate or 2-hydroxyoctanoate when compared to glycolate (Table 2 and Table 3). For the T4V/D and T5V/D mutations, various consequences were observed (Table 3). As seen when testing the glycolate oxidase activity, the k_cat_ of *At*GOX1_T4D_, *At*GOX2_T4D_ and *Zm*GO1_T5D_ was strongly reduced compared to GOX_WT_ proteins when either L-lactate or 2-hydroxyoctanoate was used (Table 2 and Table 3). In these conditions, some significant changes were observed also for certain K_M_ values but this depended on the GOX protein since only the K_M_ L-lactate of *At*GOX1_T4D_ was slightly increased (1.4-fold) while the K_M_ 2-hydroxyoctanoate of *At*GOX2_T4D_ and *Zm*GO1_T5D_ was increased by 3.5-fold and 3-fold, respectively, when compared to their corresponding GOX_WT_ proteins (Table 3). When the T4/T5 phospho-site was replaced by a valine, only *At*GOX1_T4V_ and *At*GOX2_T4V_ differed from their *At*GOX_WT_ counterparts with respect to Km 2-hydroxyoctanoate since a 2-fold decrease for *At*GOX1_T4V_ and a 2-fold increase for *At*GOX2_T4V_ was observed (Table 3). Therefore, in general, the T4/5 and T158/159 mutations led to similar effects on the calculated kinetic parameters and enzyme activities when either glycolate, L-lactate or 2-hydroxyoctanoate was used as substrate. The T4/5 and T158/159 mutations to valine did not appear to alter substrate specificity since enzymatic efficiency using either L-lactate or 2-hydroxyoctanoate was not improved except in the cases of *At*GOX1_T4V_, *At*GOX1T_158V_ and *Zm*GO1_T159V_ where k_cat_/Km ratios for the 2-hydroxyoctanoate reaction appeared to be higher.

### 2.4. Phospho-Mimetic T4/5D and T158/159D Recombinant Proteins Lack FMN

GOX activity requires FMN as a cofactor [30] and therefore a change in FMN content could explain the altered activities of our phospho-mimetic recombinant GOX proteins (Table 2 and Table 3). By measuring the ratio of the absorbance between protein (at 280 nm) and FMN (at 450 nm) (A_280nm/450nm_), it was possible to determine the presence of FMN in the purified GOX proteins used to measure enzymatic activities. An example of absorption spectra of WT and T158/159 mutated GOX proteins highlights the typical spectra associated with FMN that was missing in the GOX_T158/159D_ forms thereby indicating an absence of cofactor (Figure 1). Similar absorption spectra were carried out for each recombinant protein and used to calculate the A_280nm/450nm_ ratios given in Table 4. It can be seen that all FNR-containing GOX_WT_ proteins had a similar low A_280nm/450nm_ ratio of between 8 and 9. On the other hand, *At*GOX1_T158D_, *At*GOX2_T158D_ and *Zm*GO1_T159D_ exhibited higher A_280nm/450nm_ values of 23, 20.5 and 19.5, respectively (Table 4) since they all lacked a significant typical FMN absorption signature (Figure 1), thus indicating an absence (or an extremely reduced amount) of cofactor.

In this way, *At*GOX1_T4D_, *At*GOX2_T4D_ and *Zm*GO1_T5D_ were seen also to lack FMN since they had high A_280nm/450nm_ ratios of 28.3, 36.5 and 25.7, respectively (Table 4). Therefore, there appeared to be a good correlation between FMN content and GOX activity for these mutated forms. However, this was not always the case. *At*GOX1_T158V_, *At*GOX2_T158V_ and *Zm*GO1_T159V_ as well as *At*GOX1_T265D_, *At*GOX2_T265D_ and *Zm*GO1_T266D_ appeared to have a normal FMN content (Table 4) even though they exhibited a strong decrease of k_cat_ (Table 2). This was also seen for *At*GOX2_S212A_, *At*GOX2_S212D_, *Zm*GO1_S213A_ and *Zm*GO1_S231D_ which showed normal A_280nm/450nm_ ratios (Table 4) but reduced k_cat_ values (Table 2).

## 3. Discussion

Regulation of the photorespiratory cycle is still poorly understood even if phosphoproteomics studies have indicated that all photorespiratory enzymes except glycerate kinase can be phosphorylated [24]. In this context, GOX phosphorylation could be important since several phospho-sites have been reported and the phosphorylated residues have been conserved during the evolution of land plants (Appendix A). In this study, we explored the role of GOX phosphorylation by analyzing the kinetic parameters of both phospho-dead (to control the importance of the original residue) and phospho-mimetic recombinant Arabidopsis (C3-plant) and maize (C4-plant) photorespiratory GOX proteins.

### 3.1. Phospho-Mimetic GOX and Inhibition of Enzyme Activity

Phospho-mimetic GOX proteins exhibited different degrees of inhibition of their glycolate oxidase activity, except in one case, AtGOX1_T212D_, where the recombinant protein maintained GOX1_WT_ activity (Table 2). This inhibition was seen to correlate with reduced amounts (or the absence) of FMN (Figure 1, Table 4) for GOX_T4/5D_ and GOX_T158/158D_ but this was not observed for GOX_S212/213D_ and GOXT_265/266D_. When the phospho-mimetic mutation affected FMN content, the degree of inhibition was not constant since the mutation of T4/T5 to aspartate dramatically decreased GOX activity without changing the K_M_ glycolate, while T158D and T159D mutations resulted in inactive recombinant GOX proteins (Table 2). These modifications in both Arabidopsis and maize GOX kinetics parameters could be attributed to the lack of FMN in these mutated proteins since its presence is essential for GOX activity (Table 4) [30]. Recently, the 3D structure of apo-GOX (lacking FMN) and holo-GOX from *Nicotiana benthamiana* revealed that loop4 (residues 157–165; 156–164 of *At*GOX) and loop6 (residues 253–265; 252–264 of *At*GOX) together with a loop situated between residues 28–33 (27–32 for *At*GOX) formed a lid preventing the loss of FMN [30] (Figure 2A, Appendix A). Moreover, this lid had a strong interaction with a neighbouring GOX subunit of the tetrameric *Nb*GOX, with α-helix4 (next to loop4) forming H-bonds with E3, T5 and N6 (E2, T4 and N5 of *At*GOX) allowing a cooperative mechanism in FMN binding between GOX subunits of the same tetramer [30] (Figure 2B, Appendix A). Structural models of *Arabidopsis thaliana* GOX1 and GOX2 as well as *Zm*GO1 based on the structure of *Spinacia oleracea* GOX (PDB 1AL7) indicated that T4 formed H-bonds with R163, E165 and K169 of α-helix4 of a neighbouring GOX subunit.

When T4 was replaced by an aspartate in these models, only an H-bond with D163 of α-helix4 was present. Moreover, T158 is located close to a residue implicated in FMN-binding (T155) [31,32], the latter being also potentially phosphorylated [25]. Thus, T158 (present in loop4) and T4 (at the N-terminus) are located in essential domains for FMN binding; therefore, their mutation to aspartate, and probably their phosphorylation, may disturb FMN binding to holoGOX resulting in a less active or inactive enzyme. Furthermore, structural models indicated that when T158/159 was replaced by an aspartate, a new H-bond formed with Y129. This tyrosine residue normally formed an H-bond with FMN and when mutated to a phenylalanine (spinach GOX_Y129F_) the resulting protein had only 3.5% of GOX_WT_ activity and a k_cat_ of only 0.74 s^−1^ instead of 20 s^−1^; however, FMN content was not affected [33]. Therefore, it is possible that phosphorylation of T158/159 could alter GOX activity even if FMN was retained in the GOX protein.

The mutation of T265/266 to aspartate decreased the glycolate-dependent k_cat_ of *At*GOX1/2_T265D_ and *Zm*GO1_T266D_ compared to the wild-type recombinant protein but only changed the K_M_ glycolate of *At*GOX1_T265D_ (Table 2). As mentioned earlier, contrary to GOX_T4D/T5D_ and GOX_T158D/T159D_, the FMN content of phospho-mimetic GOX_T265D/T266D_ was comparable to recombinant GOX_WT_ protein (Table 4) even though T265/T266 is located just at the end of loop6 (residues 253–265; 252–264 of *At*GOX1/2) which is part of the “lid” structure involved in FMN loss in the pH sensor model [30]. Again, based on GOX structural models, T265 formed an H-bond with D285 of β-sheet7; however, when replaced by an aspartate this bond was broken and a new H-bond was formed with N253 of loop6 that is next to the active site H254 involved in proton abstraction during catalysis. We propose that GOX_T265/T266D_ inhibited glycolate oxidase activity by bringing about a conformational change that interfered with catalysis via the displacement of H254.

### 3.2. Phospho-Site Mutations Suggest Differences between Photorespiratory GOX Proteins

Our studies to decipher photorespiratory GOX regulation by protein phosphorylation appeared to indicate that our different GOX enzymes (*At*GOX1 versus *At*GOX2 and *At*GOX1/2 versus *Zm*GO1) did not always respond in a similar manner thus suggesting subtle differences between them even though they have a conserved photorespiratory role in planta. Both mutations at S212/S213 led to a similar decrease in the glycolate-dependent k_cat_ of *At*GOX2 and *Zm*GO1 while the k_cat_ of *At*GOX1 remained unaltered (Table 2). On the other hand, these mutations did not modify the K_M_ glycolate of the different proteins (Table 2). Therefore, while *At*GOX1 and *At*GOX2 have been shown to have a redundant photorespiratory function [7] and similar kinetic properties ([34], Table 2), these observations suggest that they could have subtle regulatory and structural differences which could potentially explain reported differences in planta [11]. Furthermore, since S212/S213 was found to be important for *At*GOX2 and *Zm*GO1 activity but not for *At*GOX1 activity, we propose that the reduced k_cat_ observed with GOX_S212/213D_ does not reflect the actual consequences of a phosphorylated GOX at S212. It is indeed difficult to explain why *At*GOX1 did not behave in a similar way to the other two GOX proteins since S212 is located in a loop between α-helixD and α-helixE that contains no known residues involved in glycolate oxidase activity. Our structural models showed that S212 was not involved in any H-bond formation; however, when replaced by an aspartate it could form a new H-bond with D54 in a loop between β-sheetA and β-sheetB of a neighbouring GOX subunit, but this was the case for all three proteins.

*At*GOX1/2 T4 and *Zm*GO1 T5 mutations to valine were done to suppress any phosphorylation but also to mimic the sequence found in both *At*HAOX1 and *At*HAOX2, two enzymes that use more efficiently long-chain hydroxyl-acids [6], and *Hs*HAOX2. As previously shown [8], *At*GOX1 and *At*GOX2 were less efficient using L-lactate as a substrate with both enzymes showing an 8-fold increase in K_M_ although the k_cat_ values were not significantly different when compared to glycolate (Table 2 and Table 3). However, contrary to the literature [8], both *At*GOX1 and *At*GOX2 were able use 2-hydroxy-octanoate as a substrate but again they were less efficient with lower k_cat_ and increased K_M_ values when compared to glycolate (Table 2 and Table 3). Interestingly, the C4-plant enzyme *Zm*GO1 had a similar k_cat_ using either L-lactate or glycolate as a substrate while displaying only a 4-fold increase of K_M_ and even though the k_cat_ with 2-hydroxy-octanoate decreased in a similar manner to *At*GOX proteins, its K_M_ 2-hydroxy-octanoate was unaltered when compared to its K_M_ glycolate (Table 2 and Table 3). Thus, even if the kinetics parameters of *Zm*GO1, *At*GOX1 and *At*GOX2 were similar for the glycolate oxidase reaction ([34], Table 2), *Zm*GO1 appeared to be able to use more efficiently both L-lactate and 2-hydroxy-octanoate (as seen from a comparison of k_ca_t/K_M_ ratios).

Differences between GOX proteins with respect to 2-hydroxy-octanoate were also observed when comparing the effect of T4/5 and T158/159 mutations to valine. Indeed *At*GOX1_T4V_, *At*GOX2_T4V_ and ZmGO1T5V each exhibited a different alteration of K_M_ 2-hydroxy-octanoate compared to their GOX_WT_ counterparts (Table 3). This amino acid being important for GOX quaternary structure [30], its replacement by a valine may modify 2-hydroxy-octanoate specificity towards that of *At*HAOX1 and *At*HAOX2 due to possible subtle differences in 3D structure.

T158V and T159V mutations induced the same modifications of K_M_ and k_cat_ when using either of the three substrates tested thus indicating that the mutation affected the global enzymatic mechanism and it did not appear to be involved in determining substrate specificity (Table 2 and Table 3). Finally, *At*GOX1/2_T158V_ and *Zm*GO1_T159v_ proteins were poorly active whatever the substrate used however for each protein the K_M_ appeared to be lower than their GOX_WT_ protein counterparts (Table 2 and Table 3). This inferred that T158 was involved in substrate binding as well as enzymatic activity and that when mutated to valine (as in Arabidopsis HAOX proteins) this somehow improved substrate binding. In conclusion to this part of the discussion, we can say that V4/5 and V157/158 containing GOX proteins were not transformed into HAOX enzymes as they were unable to use 2-hydroxy-octanoate more efficiently.

### 3.3. Discovering Conditions Inducing GOX Phosphorylation and Identifying GOX Kinases

To date, the only evidence of GOX protein phosphorylation has come from phosphoproteomics studies using mass spectroscopy analyses. From Table 1 it can be seen that only the peptide containing phosphorylated T158 has been reported more than once and although the accuracy of the methods in this domain have increased over the years, there are still a number of technical constraints and peptide identification is prone to error. It is therefore important to confirm GOX phosphorylation using other methods and/or confirm by mass spectroscopy using less complex samples instead of total protein extracts. Furthermore, it is necessary to identify conditions that bring about GOX phosphorylation so as to better understand its function. The T4 phospho-site of *At*GOX1 and *At*GOX2 was identified in leaves of plants subjected to oxygen depletion (5% of O_2_ for 3 h) (Table 1). In such low O_2_ conditions, photorespiration would be less important and GOX activity could be reduced via T4/T5 phosphorylation. Although the T158 phospho-site has been seen in several different phosphoproteome experiments, no significant differences in the quantity of this phosphopeptide were reported whatever the conditions tested (Table 1), even when comparing dark versus light and low versus high CO_2_ concentrations that are expected to modulate photorespiratory activity [28]. Thus, T158 phosphorylation may have a role that is not linked to photorespiration. Several studies have implicated GOX in response to pathogen attack [9,35,36], and therefore regulation of GOX activity, together with catalase, may be a way to modulate H_2_O_2_ production as part of plant defence signalling and this could involve protein phosphorylation. Arabidopsis phospho-site S212 was identified in a phosphoproteomic study of the triple kinase mutant *snrk2.2/2.3/2.6* subjected either to ABA or dehydration treatments [26]. SnRK2.2, SnRK2.3 and SnRK2.6 (also known as OST1) are three kinases activated by ABA [37] and we have recently shown that another photorespiratory enzyme (SHMT1) responded to ABA and altered stomatal movements in response to salt stress [38]. Thus, GOX could be a target of leaf ABA signalling either in mesophyll cells or stomata. Indeed, the quantity of the TL(pS)WK phosphopeptide was shown to increase in response to either ABA or dehydration and this increase was absent in *snrk2.2/2.3/2.6* seedlings [26].

Therefore, phosphoproteomics studies have had very limited success in providing insights into the eventual role of GOX phosphorylation. Since this post translational modification is a rapid and reversible response to environmental stimuli, experiments should be conducted to identify when GOX is differentially phosphorylated. With respect to GOX photorespiratory function, different conditions expected to modulate photorespiratory flux should be examined such as day/night cycle, variable CO_2_/O_2_, and stresses like heat, drought, salt and high light as well as pathogen attack. To simplify the detection of GOX phosphopeptides by mass spectroscopy, GOX should be specifically immunoprecipitated from soluble protein extracts at different times during the stress treatments or during the day/night cycle. In this way, it should be possible to identify conditions where GOX phosphorylation is modulated.

Of course, our in vitro data using recombinant phospho-mimetic GOX proteins only give an indication of what GOX phosphorylation might actually be doing with respect to enzyme activity. Indeed, GOX was produced in *Escherichia coli* as a recombinant phospho-mimetic protein and therefore we do not know whether this led to a non-physiological alteration of GOX structure (and activity). Since the modification was constitutive and present as soon as the protein was synthesized, it does not reflect a reversible phosphorylation of a protein. For instance, is the absence of FMN in GOX_T4/5D_ and GOX_T158/159D_ due to structural changes induced by the aspartate that inhibits FMN entry into the apo-protein to form the halo-protein, and therefore would phosphorylation lead to changes that favour FMN removal from the halo-protein? Furthermore, phospho-mimetic mutations do not always lead to the modifications in protein function brought about by an actual phosphorylation. Thus, to assess the real role of GOX phosphorylation, identification of the protein kinases (and also the phosphatases) involved would be crucial. Based on the SUBA database (http://suba.plantenergy.uwa.edu.au/), 33 protein kinases are predicted to be addressed to the peroxisome. Using machine learning methods to identify proteins carrying plant peroxisomal PST1 targeting signals, 11 protein kinases were identified as being addressed to the peroxisome [39]. More recently, a list of about 200 confirmed Arabidopsis peroxisomal proteins was compiled and four kinases and eight phosphatases or phosphatase subunits were identified [40]. A strategy to identify *At*GOX1/2 kinases could be to retrieve knock-out mutant lines for predicted and/or verified peroxisomal protein kinases and compare *At*GOX1/2 phosphorylation status by mass spectroscopy after *At*GOX1/2 immunoprecipitation from soluble proteins extracted from wild-type and mutant plants treated to previously identified conditions that induce *At*GOX1/2 phosphorylation.

In conclusion, photorespiratory GOX has the potential to be regulated by protein phosphorylation at several distinct sites. In general, phospho-mimetic recombinant GOX proteins exhibited reduced activities and this could be explained by predicted changes in structural interactions affecting key residues involved in FMN binding and catalysis. Phospho-mimetic GOX did not show any alteration in substrate specificity although C4-plant *Zm*GO1 did appear to have a relaxed substrate specificity when compared to C3-plant *At*GOX1 and *At*GOX2.

## 4. Materials and Methods

### 4.1. Plasmid Constructions and Site-Directed Mutagenesis

To produce recombinant proteins, previously made pET28a-*At*GOX1, pET28a-*At*GOX2 and pET28a-*Zm*GO1 expression plasmids [34] were used as templates to introduce point mutations using specific primers pairs (Appendix A) and the QuikChange^®^ II XL site-directed mutagenesis kit (Agilent^®^, Les Ulis, France), according to the manufacturer’s instructions. This strategy generated T4V, T4D, T158V, T158D, S212A, S212D, T265A, T265D (*At*GOX1 and *At*GOX2) and T5V, T5D, T159V, T159D, S213A, S213D, T266A, T266D (*Zm*GO1) mutated proteins. All constructions were subsequently verified by DNA sequencing using T7 and T7-term primers (Appendix A).

### 4.2. Purification of Recombinant GOX Proteins, SDS-PAGE and Determination of FMN Content

GOX proteins were purified as described by Dellero et al. [34]. The purity of each recombinant GOX protein was checked by SDS-PAGE (10% acrylamide) stained with Coomassie Brilliant Blue [41]. Recombinant GOX proteins used to measure FMN content were purified without FMN in buffers and FMN levels were evaluated by spectrophotometry using the A_280_/A_450 nm_ ratio.

### 4.3. GOX Activity Measurements

Enzyme activities were measured using 5 µg of purified recombinant GOX in 50 mM Tris-HCl, 0.1 mM FMN, pH 8.0 and different glycolate (0.05 to 10 mM), L-lactate (0.3 to 10 mM) and 2-hydroxy-octanoate (0.15 to 5 mM) concentrations by an enzyme-coupled reaction at 30 °C. H_2_O_2_ produced by GOX activity was quantified in the presence of 0.4 mM *o*-dianisidine and 2 U horseradish peroxidase by measuring the ∆A_440nm_ using a Varian Cary 50 spectrophotometer. K_M_ and k_cat_ values were calculated using SigmaPlot 13.0 software, based on the curve fitting Michaelis–Menten equation: v_0_ = V_max_[S]/(K_M_ + [S]).

### 4.4. Structural Analysis

*At*GOX1, *At*GOX2 and *Zm*GO1 3D-structures were designed based on the spinach GOX (UniProtKB: P05414) 3D-structure (PDB: 1al7.1) using the SWISS-MODEL server (https://swissmodel.expasy.org/) modelling service. Manipulation of 3D-structures was realized using Deepview (Swiss Pdb-viewer) (https://spdbv.vital-it.ch/) and included the compute H-bond formation and mutation tools.

## Figures and Tables

**Figure 1 plants-09-00027-f001:**
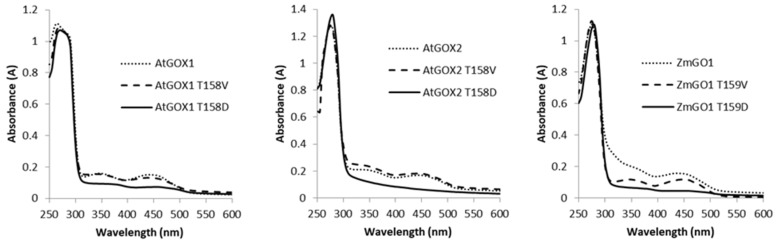
Absorption spectra of recombinant *At*GOX1, *At*GOX2 and *Zm*GO1 and their T158/T159 phospho-site mutated forms.

**Figure 2 plants-09-00027-f002:**
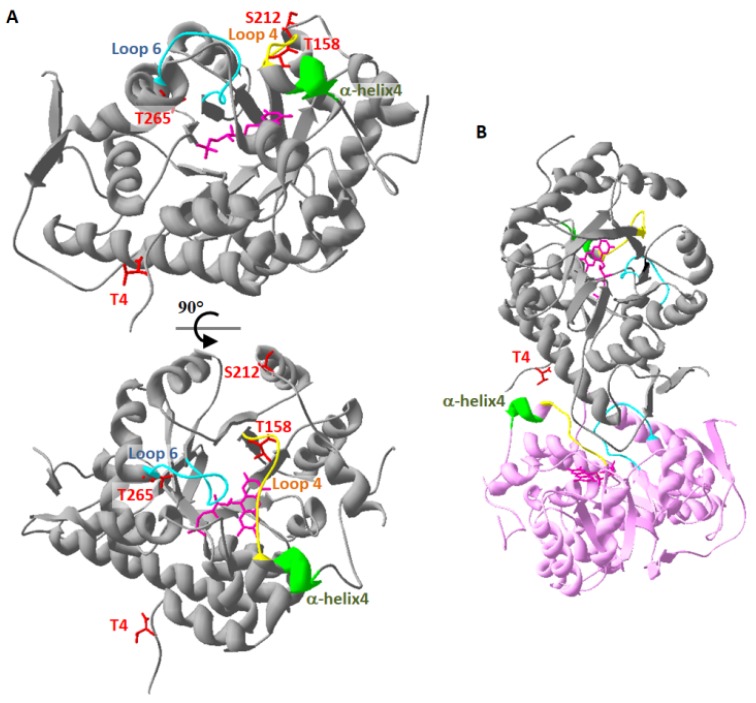
Localisation of phosphorylated residues and important structures of *At*GOX1. A structural model of *At*GOX1 based on the 3D-structure of spinach GOX [31]. (**A**) *At*GOX1 monomer showing phosphorylated residues T4, T158, S212 and T265 (red), loops 4 (yellow) and 6 (blue), α-helix4 (green) and the flavin mononucleotide (FMN) (pink). The bottom structure is rotated by 90° with respect to the top structure. (**B**) View of two *At*GOX monomers (in grey and in pink) and the proximity between T4 (red) of a GOX subunit with α-helix4 (green) of a neighbouring GOX subunit.

**Table 1 plants-09-00027-t001:** Phosphopeptides associated with *At*GOX1 and *At*GOX2.

Gene Name, Locus	Phosphopeptide	Peptide Location	Sample Type, Age	Treatment	References
***AtGOX1, At3g14420 orAtGOX2, At3g14415***	MEI(**pT_4_**)NVTEYDAIAK	1–14/367	Leaves	Oxygen depletion	PhosPhAt 4.0 ^1^
	AIAL(**pT_155_**)VDTPRL	151-161/367	Seedlings, 11 days	Sucrose depletion	[25]
	AIALTVD(**pT_158_**)PRL	151-161/367	Seedling, 2 weeks	ABA and dehydration	[26]
			Seedlings, 10 days	Continuous light for 24 h	[27]
			Rosette	Varying O_2_/CO_2_ conditions	[28]
	TL(**pS_212_**)WK	210-214/367	Seedlings, 2 weeks	ABA and dehydration	[26]
	QLDYVPA(**pT_265_**)ISALEEVVK	258-273/367	Cauline leaves, 2 months		[29]
***AtGOX1 At3g14420***	NHI(**pT_355_**)TEWDTPR	352-362/367	Seedlings, 10 days	Continuous light for 24 h	[27]

The phosphorylated residue in each peptide is shown in bold with its position numbered according to *At*GOX1. Peptide location shows the position of each phosphopeptide with respect to the number of residues in *At*GOX1. ^1^ PhosPhAt 4.0: http://phosphat.mpimp-golm.mpg.de/.

**Table 2 plants-09-00027-t002:** Effect of phospho-site mutations on glycolate-dependent kinetic parameters of recombinant *At*GOX1, *At*GOX2 and *Zm*GO1.

	K_M_	k_cat_		K_M_	k_cat_		K_M_	k_cat_
*At*GOX1	(µM)	(s^−1^)	*At*GOX2	(µM)	(s^−1^)	*Zm*GO1	(µM)	(s^−1^)
WT	210 ± 90	11.12 ± 3.08	WT	279 ± 30	10.93 ± 3.48	WT	126 ± 34	11.45 ± 1.86
T4V	123 ± 35	9.59 ± 1.57	T4V	251 ± 70	12.09 ± 1.50	T5V	135 ± 56	8.67 ± 2.32
T4D	151 ± 44	**0.61 ± 0.46 ***	T4D	390 ± 215	**1.40 ± 0.42 ***	T5D	157 ± 31	0.53 ± 0.22 *
T158V	**100 ± 19 ***	**2.36 ± 0.47 ***	T158V	**133 ± 83 ***	**2.49 ± 0.60 ***	T159V	**46 ± 20 ***	**2.40 ± 0.35 ***
T158D	no activity		T158D	no activity		T159D	no activity	
S212A	257 ± 31	12.13 ± 2.15	S212A	249 ± 14	**6.10 ± 0.64 ***	S213A	89 ± 13	**7.19 ± 0.46 ***
S212D	277 ± 24	11.46 ± 2.31	S212D	325 ± 13	**5.23 ± 0.41 ***	S213D	129 ± 28	**6.39 ± 1.45 ***
T265A	276 ± 112	14.02 ± 2.65	T265A	237 ± 67	11.33 ± 3.27	T266A	91 ± 18	11.75 ± 2.94
T265D	**531 ± 50 ***	**0.14 ± 0.06 ***	T265D	388 ± 159	**2.06 ± 1.09 ***	T266D	173 ± 65	**0.62 ± 0.77 ***

Mean values ± SD from three independent biological replicates. Statistical significance was determined by a Student’s *t*-test. Values in bold and marked by an asterisk were significantly different compared to the corresponding WT protein (*p* < 0.05).

**Table 3 plants-09-00027-t003:** Effect of selected phospho-site mutations on L-lactate and 2-hydroxy-octanoate dependent kinetic parameters of recombinant *At*GOX1, *At*GOX2 and *Zm*GO1.

	L-lactate		2-hydroxy-octanoate	
Enzyme	K_M_	k_cat_	K_M_	k_cat_
	(µM)	(s^−1^)	(µM)	(s^−1^)
*At*GOX1_WT_	1664 ± 293	9.09 ± 1.37	757 ± 82	5.95 ± 0.58
*At*GOX1_T4V_	1844 ± 911	9.01 ± 3.59	**350 ± 221 ***	4.66 ± 1.56
*At*GOX1_T4D_	**2377 ± 270 ***	**0.30 ± 0.07 ***	1255 ± 818	**0.39 ± 0.17 ***
*At*GOX1_T158V_	**549 ± 160 ***	**2.30 ± 0.56 ***	**191 ± 51 ***	**2.89 ± 0.89 ***
*At*GOX1_T158D_	no activity		no activity	
*At*GOX2 _WT_	2094 ± 791	6.81 ± 1.54	487 ± 99	3.78 ± 1.26
*At*GOX2 _T4V_	2119 ± 453	6.65 ± 1.17	**951 ± 91 ***	4.05 ± 2.94
*At*GOX2 _T4D_	2019 ± 169	**0.67 ± 0.17 ***	**1721 ± 677 ***	**0.59 ± 0.30 ***
*At*GOX2 _T158V_	**439 ± 171 ***	**1.71 ± 0.53 ***	501 ± 116	2.46 ± 0.83
*At*GOX2 _T158D_	no activity		no activity	
*Zm*GO1 _WT_	495 ± 75	10.85 ± 0.08	136 ± 19	5.89 ± 0.14
*Zm*GO1 _T4V_	488 ± 233	11.99 ± 1.68	168 ± 59	5.88 ± 1.59
*Zm*GO1 _T4D_	719 ± 233	**0.57 ± 0.26 ***	**414 ± 84 ***	**0.53 ± 0.40 ***
*Zm*GO1 _T158V_	**161 ± 29 ***	**3.55 ± 0.53 ***	**48 ± 36 ***	**3.22 ± 0.39 ***
*Zm*GO1 _T158D_	no activity		no activity	

Mean values ± SD from three independent biological replicates. Statistical significance was determined by a Student’s *t*-test. Values in bold and marked by an asterisk were significantly different compared to the corresponding WT protein (*p* < 0.05).

**Table 4 plants-09-00027-t004:** A_280/450nm_ ratios of recombinant *At*GOX1, *At*GOX2 and *Zm*GO1 proteins.

*At*GOX1	Ratio 280/450 nm	*At*GOX2	Ratio 280/450 nm	*Zm*GO1	Ratio 280/450 nm
WT	8.7 ± 2.1	WT	8.3 ± 0.7	WT	8.1 ± 1.8
T4V	10.7 ± 4.1	T4V	6.9 ± 2.2	T5V	6.7 ± 3.0
T4D	**28.3 ± 6.1 ***	T4D	**36.5 ± 7.3 ***	T5D	**25.7 ± 6.2 ***
T158V	8.7 ± 2.3	T158V	8.2 ± 1.7	T159V	7.9 ± 1.1
T158D	**23.0 ± 8.3 ***	T158D	**20.5 ± 9.3 ***	T159D	**19.5 ± 7.1 ***
S212A	6.9 ± 3.3	S212A	12.4 ± 5.1	S213A	6.6 ± 3.5
S212D	6.6 ± 4.0	S212D	10.4 ± 2.2	S213D	6.1 ± 1.7
T265A	9 ± 3.3	T265A	9.6 ± 2.4	T266A	7.0 ± 0.1
T265D	8.5 ± 2.5	T265D	11.8 ± 4.5	T266D	21.2 ± 12.1

Mean values ± SD from three independent experiments. Statistical significance was determined by a Student’s *t*-test. Values in bold and marked by an asterisk were significantly different compared to the corresponding WT protein (*p* < 0.05).

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
