# Peer review of "Enzymatic Properties of Recombinant Phospho-Mimetic Photorespiratory Glycolate Oxidases from Arabidopsis thaliana and Zea mays"

_plants, 2019, doi:10.3390/plants9010027_

Round 1
Reviewer 1 Report
The manuscript by Jossier et al. aims to study the regulation of photorespiration and particularly gycolate oxidades by phosphorylation. The experiments are correct, the results are clear and provide exhaustive information about GOX regulation. They nicely discuss their results comparing them with structural models.
Abstract.There must be a mistake in the numbering of the phosphorylation sites. T168 and T158 are the same or different phosphorylation sites? I guess T158 is the correct one.
Table 1 needs to be clarified. It is not specified what 1-14/367 means. I guess the first numbers are the peptide location, and 367 is the total number of animo acids in the protein. For clarity the phosphorylated residues should also be numbered in the table as described in the results section. The biological relevance of their findings is also discussed.
In tables 2,3 it should be specified whether three independent experiments refers to biological replicates or replicates from the same extract.
Discussion could be shortened.
Author Response
DONE Abstract.There must be a mistake in the numbering of the phosphorylation sites. T168 and T158 are the same or different phosphorylation sites? I guess T158 is the correct one.
DONE Table 1 needs to be clarified. It is not specified what 1-14/367 means. I guess the first numbers are the peptide location, and 367 is the total number of animo acids in the protein. For clarity the phosphorylated residues should also be numbered in the table as described in the results section.
DO NOT UNDERSTAND THIS COMMENT. IS IT A STATEMENT OR IS IT A REQUEST? The biological relevance of their findings is also discussed.
DONE In tables 2,3 it should be specified whether three independent experiments refers to biological replicates or replicates from the same extract.
PLEASE SUGGEST WHERE? ALTHOUGH WE PREFER NOT TO SHORTEN THE DISCUSSION Discussion could be shortened.
Reviewer 2 Report
In the manuscript “Enzymatic properties of recombinant phospho-mimetic photorespiratory glycolate oxidases from Arabidopsis thaliana and Zea mays” the authors Mathieu Jossier, Yanpei Liu, Sophie Massot and Michael Hodges use phosphorylation analysis studies to assess the question of photorespiration regulation. By mutating selected phosphorylation sites of Arabidopsis and maize GOX proteins they compare their kinetic and substrate specificity. The experiments are set up very well and thoroughly. The discussion is also very thorough and I certainly agree with the proposed further experiments in the discussion. Besides the request for the following changes I find no further faults in the manuscript:
Abstract:
The abstract is a bit sloppy written, containing some mistakes. Please change:
Line 18: T168 should be T158
Line 24: S265A should be T265
The last sentence (Lanes 26-27, talking about substrate specificity) has no basis in the abstract. Additional sentence should be added to the abstract to corroborate this final statement.
Main text:
Line 51: remove second AtGOX3
Line 53: remove “AtHAOX proteins”
Line 54: Knock-out
Line 65: I am not familiar with the meaning of “growth in air”, would the authors be so kind to clarify this expression for me (and for other future readers)?
Lines 78 to 88 should be removed to result section. The results from this research should not be mentioned in the introduction. Only the last sentence (Line 88-91) should remain here.
Line 148: It is the other way round – 2-fold decrease of Km glycolate, kcat was not changed!
(Table 2: S213A and S213D in ZmGO1 – the difference in kcat doesn’t seem significant)
Line 153: “Values in bold…” – there are no values in bold
Figure2: The quality is poor. The figure MUST BE IMPROVED. T4, T158, S212 and T265 is not written visibly. Also, in figure legend (Line 263) instead of loops 4 (blue) and 6 (yellow) should be written: loops 4 (yellow) and 6 (blue)
References:
Line 451: Remove doi
Line 494: The Journal is missing
Line 496: The Journal is missing again
Line 532: Add dot after Peer J
Line 538: Remove (9)
Figure S3. Should be divided into subfigures (A to E), properly formatted and detailed descriptions/legends added.
Author Response
Abstract:
CHANGED Line 18: T168 should be T158 / Line 24: S265A should be T265
REMOVED SUBSTRATE SPECIFITY FROM SENTENCE The last sentence (Lanes 26-27, talking about substrate specificity) has no basis in the abstract.
Main text:
DONE Line 51: remove second AtGOX3 /Line 53: remove “AtHAOX proteins” / Line 54: Knock-out
THIS MEANS GROWTH IN AIR (400 PPM CO2) AS APPOSED TO GROWTH IN CO2-ENRICHED AIR (3000 PPM CO2) - WE HAVE ADDED (400 PPM CO2 IN BRACKETS) Line 65: I am not familiar with the meaning of “growth in air”, would the authors be so kind to clarify this expression for me (and for other future readers)?
DONE Lines 78 to 88 should be removed to result section.
WE BELIEVE THE ORIGINAL TEXT WAS CORRECT Line 148: It is the other way round – 2-fold decrease of Km glycolate, kcat was not changed!
THE SIGNIFICANT DIFFERENCES ARE ALWAYS WITH RESPECT TO THE WT GOX (Table 2: S213A and S213D in ZmGO1 – the difference in kcat doesn’t seem significant)
ADDED THE BOLD Line 153: “Values in bold…” – there are no values in bold
WE HAVE TRIED TO IMPROVE THIS FIGURE Figure2: The quality is poor. The figure MUST BE IMPROVED.
References:
ALL CORRECTED - sorry - we should not trust Mendeley for this! Line 451: Remove doi /Line 494: The Journal is missing /Line 496: The Journal is missing again/ Line 532: Add dot after Peer J /Line 538: Remove (9)
DONE Figure S3. Should be divided into subfigures (A to E), properly formatted and detailed descriptions/legends added.
Reviewer 3 Report
See file

Author Response
DONE IN NEW SUPP FIG 2 Please include a figure or table showing an alignment of the phosphorylated amino acids of GOX proteins used in this study and the long-chain Hydroxyl acids and lactate oxidases. This alignment could also include amino acids showing to be responsible for substrate binding (Hackenberg et al. 2011, Esser et al. 2014,)
THIS PAPER WAS NOT MENTIONED FOR AtGOX2 SINCE WE BELIEVE THE KmS WERE INVERTED (based on our data and that of another paper). PERHAPS THE REVIEWER COULD GIVE US HIS OPINION? I missed a well investigated comparison of kinetic parameters of different GOX proteins from plants (AtGOX2 for example was already published by Hackenberg et al. 2011) and algae.
IF EACH ASSAY WORKS CORRECTLY IT SHOULD NOT IMPACT THE KINETIC PARAMETERS Especially in the background of different strategies to overcome the oxygenase reaction of Rubisco and also the different enzyme assays used by other groups (coupled enzyme assay vs oxygen consumption vs alternative electron acceptors). WE ARE NOT SURE WHAT YOU WANT TO SAY HERE.
The authors address the regulation of GOX activity by phosphorylation. Therefore they use an in vitro assay with a recombinant enzyme expressed in coli, which is a prokaryotic expression system. Do the prokaryotic kinases phosphorylate the same amino acids as the plant kinases do? THE RECOMBINANT GOX IS NOT PHOSPHORYLATED IN E. COLI - THIS WAS CHECKED BY MASS SPECTROMETRY
I missed the proof of evidence for the phosphorylation of the recombinant GOX enzymes (including also the phospho-mimetic mutants). OUR RECOMBINANT PROTEINS WERE NOT PHOSPHORYLATED - THEY ARE JUST PHOSPHO-MIMETIC
Is the inactivation or activity reduction of GOX a result of phosphorylation or the kind of amino acid which was substituted? THE EFFECT IS DUE TO REPLACING THE PHOSPHORESIDUE BY A PHOSPHOMIMETIC ASPARTATE The authors may performed an in vitro dephosphorylation of the GOX variants to confirm their hypothesis of regulation. SINCE OUR PROTEINS ARE NOT PHOSPHORYLATED THIS IS NOT POSSIBLE
The GOX isoforms in Arabidopsis (and also rice, see Zhang et al. 2017) prefer different substrates and therefore exhibit different functions in the plant metabolism. However, they were also expressed in different plant organs and developmental stages (see Esser et al. 2014 and Engqvist et al.). Why should their substrate specifity be regulated by post-translational phosphorylation when for instance Arabidopsis has up to 5 isoforms? WE CHECKED WHETHER TWO PHOSPHO-MIMETIC MUTATIONS DIFFERENTIALLY MODIFIED A SPECIFIC ACTIVITY (GLYCOLATE/LACTATE/FATTY ACID) BECAUSE PERHAPS A CERTAIN SUBSTRATE BINDING MIGHT HAVE BEEN AFFECTED MORE COMPARED TO THE OTHER SUBSTRATES TESTED
The authors propose a shift in substrate specifity of GOX regulated by phosphorylation. Is there a well-studied example from another enzyme? WE DID NOT PROPOSE A SHIFT, WE TESTED THE IDEA AND IT APPEARS NOT TO BE THE CASE ALTHOUGH NOT ALL OF THE 3 TESTED GOX PROTEINS WERE AFFECTED IN A SIMILAR MANNER.
NO WE DO NOT KNOW OF A WELL STUDIED EXAMPLE FOR AN ENZYME ALTHOUGH KINASES CAN HAVE THEIR SUBSTRATES ALTERED BY PHOSPHORYLATION
Why do the authors measure the activity with l-lactate and 2-hydroxy-octonoate and not with glyoxylate or other fatty acids? WE CHOSE 2-HYDROXY-OCTONATE BECAUSE IT IS A GOOD SUBSTRATE FOR BOTH ARABIDOPSIS HAOXs (ESSER ET AL 2014) & WE DID NOT LOOK AT GLYOXYLATE BECAUSE IT WAS FOUND NOT WORK WITH AtGOX2 (ESSERET AL 2014).
The authors analysed affinity for l-lactate. Therefore, I missed the comparison of AtGOX1 amino acid sequence with other biochemically confirmed lactate oxidases. For instance, the tyrosin at the 4th aa residue in AtGOX1 and 2 correspond to a valine in the LOX of Aerococcus (BAA09172). Which is in fact the amino acid used for mimicing a phospho-dead GOX protein. WE BELIEVE THE REVIEWER MEANS THREONINE - INDEED SEQUENCE ALIGNMENTS PLACE A VALINE AT THE POSITION OF THR4 IN THIS LOX - BUT IT IS NOT CONSERVED IN LOXs (SEE NEW SUPP FIG 2).
The authors mention that the regulation of photorespiratory enzymes is poorly documented. This might be true for post-translation phosphorylation. However, there is a newly published review summing up hudge post-translational regulation sites of even the photorespiratory proteins transported to the peroxisomes (Sandalio et al. 2019). Moreover, it was shown that the maize glycerate kinase is redox regulated whereas the glycerate kinase of Arabidopsis is not (Bartsch et al. 2010), which could be well used to better discuss the use of the maize GOX in comparison to Arabidopsis. In addition to post-translation regulation photorespiratory enzymes are also regulated by its metabolites (see e. g. Timm et al. 2013). WE HAVE ADDED THESE REFS TO THE INTRODUCTION & CHANGED "DOCUMENTED" BY "UNDERSTOOD"
The authors should focused on the following points: different amino acids used for substitution and the corresponding structural changes postulated. Comparison of a C3 vs C4 GOX protein, why do you expect they are different? WE DID NOT EXPECT ANYTHING - WE HAVE ALREADY PUBLISHED A WORK IN JBC SHOWING THAT C3 & C4 GOX ARE VERY SIMILAR FOR THE GOX REACTION.
The authors should focused on the following points: WE BELIEVE THAT OUR DISCUSSION IS FOCUSED ON THESE 3 POINTS WITH AN ADDITIONAL PART TO DISCUSS PUBLISHED PHOSPHOPROTEOMICS DATA AND FUTURE WORK TO BETTER UNDERSTAND PHOSPHORYLATION OF GOX.
The phosphorylation of the recombinant GOX proteins was not proofed. In addition, nothing is known about the kinase phosphorylating the enzymes in vivo. Therefore, the regulatory aspect should be discussed with more caution and as only a site aspect.WE BELIEVE THAT THIS WAS ONLY BRIEFLY DISCUSSED AND BASED ON THE PUBLISHED PHOSPHOPROTEOMICS DATA
In addition there are some minor points have to be changed: ALL CHANGED EXCEPT Table 2 and 3: summarize them to one big table WE DID NOT DO THISAS WE BELIEVE THAT IT IS CLEARER WITH 2 TABLES
Round 2
Reviewer 3 Report
THIS PAPER WAS NOT MENTIONED FOR AtGOX2 SINCE WE BELIEVE THE KmS WERE INVERTED (based on our data and that of another paper). PERHAPS THE REVIEWER COULD GIVE US HIS OPINION?
I have no idea, thats why I asked for comparison of the enzyme assays used. In fact, three different tests were used giving different results which should not be. However, they occur. I made the observation, that the maximum peak of the enzyme activity is later for GOX compared to LOX. I did not checked if this is f.e. allosteric effect.
THE RECOMBINANT GOX IS NOT PHOSPHORYLATED IN E. COLI - THIS WAS CHECKED BY MASS SPECTROMETRY
Have you state this in the manusscript? I could not find it and it is important to know! And it makes it more important to check the enzyme activity in vivo. I know that this is out of scope for this manuscript.
In addition there are some minor points have to be changed: ALL CHANGED EXCEPT Table 2 and 3: summarize them to one big table WE DID NOT DO THISAS WE BELIEVE THAT IT IS CLEARER WITH 2 TABLES
I would really like to see only one table and avoiding scrolling up and down to compare all values with each other. Especially, if there is no nice graph summarizing the results. However, you are the authors